# Efficient Transformer Attention for SNNs via Hadamard Simplification

Tingting Jiang [* 1]   Jiangrong Shen [* 2 3]   Long Chen [4]   Yaxin Li [5]   Qi Xu [1]

## Abstract

Spiking Neural Networks (SNNs) enable low-power, event-driven computation, but Transformer-based SNNs remain difficult to deploy on neuromorphic hardware due to dense operations and communication overhead. We propose two simplified attention mechanisms, **Simplified Spiking Attention (SSA)** and **Ultra-Simplified Spiking Attention (USSA)**, which replace matrix multiplication with Hadamard products and eliminate hardware-unfriendly components such as multi-head attention and scaling. We show that consecutive masking is redundant and analyze a spiking-order effect in which early spiking contributes more temporal information to attention modulation. On CIFAR-10, CIFAR-100, and DVS-Gesture, SSA achieves accuracies of 96.38%, 79.45%, and 97.56%, respectively, while reducing computational complexity from $\mathcal{O}(N^2D)$ to $\mathcal{O}(ND)$ and communication complexity from $\mathcal{O}(N^2)$ to $\mathcal{O}(ND)$. USSA further reduces communication complexity to $\mathcal{O}(N)$ with only marginal accuracy degradation. On ImageNet-1K, SSA and USSA achieve 76.91% and 77.27% accuracy, respectively, demonstrating scalability to large-scale classification.

## 1. Introduction

Neuromorphic computing (Indiveri & Liu, 2015; Davies et al., 2018) enables energy-efficient, brain-inspired edge intelligence. Among its algorithmic models, spiking neural networks (SNNs) (Maass, 1997) are particularly attractive due to their biological plausibility and event-driven computation, aligning naturally with low-power neuromorphic hardware. However, traditional SNNs often lack representational capacity and training efficiency for complex perception and cognitive tasks. To address these limitations, recent work draws inspiration from advanced ANN architectures, especially Transformers (Vaswani et al., 2017), which excel at modeling global dependencies (Zhou et al., 2023; Shi et al., 2024). Transformer-based SNNs incorporating self-attention have demonstrated improved performance in both static and event-based vision tasks.

Despite these advances (Zhou et al., 2023; Yao et al., 2023a; Hua et al., 2025; Shi et al., 2024; Zhou et al., 2024), deployment on neuromorphic hardware remains challenging. ANN-inspired designs often rely on dense operations—matrix multiplication, multi-head attention, tensor reshaping, scaling factors, and explicit patch embedding—whose computational and memory-access patterns conflict with the sparse, event-driven, locality-centric nature of neuromorphic systems. As a result, these models incur substantial communication overhead and energy consumption (Fang et al., 2024; Huang et al., 2023), limiting scalability. Such costs, often neglected in theoretical analyses, dominate practical energy usage (Wang et al., 2024), preventing theoretical gains from translating into system-level benefits.

To bridge this "algorithm–hardware" gap, we propose two hardware-efficient attention mechanisms within the SpikingResformer framework (Shi et al., 2024): **S**implified **S**piking **A**ttention (SSA) and **U**ltra-**S**implified **S**piking **A**ttention (USSA), both based on the Hadamard product. These mechanisms replace communication-intensive Transformer operators—such as matrix transposition, multi-head attention, scaling, and explicit patch embedding—substantially reducing computational and communication complexity. Information-theoretic analysis and experiments reveal redundancy in consecutive Hadamard-product masking operations, enabling simplification to a single effective mask without performance loss. Additionally, we investigate the effect of spike timing, showing that early-spiking neurons preserve richer temporal information. Our core contributions are summarized as follows:

- We   propose   two   novel   attention   mecha-

---

[1]School of Computer Science and Technology, Dalian University of Technology, Dalian, China [2]School of Computer Science and Technology, Xi'an Jiaotong University, Xi'an, China [3]National Key Laboratory of Human-Machine Hybrid Augmented Intelligence, Xi'an, China [4]Faculty of Medicine, Imperial College London, London, United Kingdom [5]School of Control Science and Engineering, Dalian University of Technology, Dalian, China. Correspondence to: Qi Xu <xuqi@dlut.edu.cn>.

*Proceedings of the $43^{rd}$ International Conference on Machine Learning*, Seoul, South Korea. PMLR 306, 2026. Copyright 2026 by the author(s).

nisms—Simplified Spiking Attention (SSA) and Ultra-Simplified Spiking Attention (USSA)—that replace communication-intensive matrix multiplication with Hadamard products, significantly improving deployability on neuromorphic hardware.

- Through theoretical analysis and experiments, we demonstrate that early-spiking neurons preserve richer temporal information in Hadamard-based attention mechanisms, revealing the spiking-order effect in temporal modeling.

- We show that consecutive masking operations are redundant and that a single-mask design achieves higher accuracy with lower computational cost.

## 2. Related Works

### 2.1. Transformer-Based SNNs for Accuracy Gains

The integration of Transformer architectures into SNNs has emerged as a primary pathway for enhancing model capacity. Pioneering works like Spikeformer (Zhou et al., 2023) and Spiking Transformer (Zhou et al., 2026) demonstrated that self-attention mechanisms could be successfully adapted to the spiking domain, achieving significant accuracy gains on image and event-based tasks. Subsequent innovations, such as Spike-driven Transformer (Yao et al., 2023a) and QKFormer (Zhou et al., 2024), further explored efficient attention variants by replacing dense matrix multiplication with Hadamard products. However, these advances remain largely confined within an ANN-transplant paradigm. They predominantly retain the fundamental structure of standard Transformers—including multi-head attention, tensor transposition and reshape, scaling factors, and explicit patch embedding—which, as noted in the introduction, are inherently misaligned with neuromorphic hardware constraints. Consequently, this line of work has primarily addressed the accuracy facet of the problem, while leaving the critical deployment efficiency challenge unresolved.

### 2.2. The Gap in Hardware-Aware Attention Design

In parallel, extensive efforts have been made to improve the hardware efficiency and deployability of spiking neural networks on neuromorphic platforms through sparse computation, structural simplification, and temporal reduction. For example, Shi et al. (Shi et al., 2024) propose SpikingResformer by adopting convolutional or ResNet-style backbones; however, its self-attention module still relies on dense matrix multiplication inherited from ANN-style Transformers, which can introduce considerable communication overhead on neuromorphic hardware. Other approaches apply efficiency-oriented techniques such as quantization (Qiu et al., 2025), pruning (Liu et al., 2024), or time-step reduction (Song et al., 2024). These methods improve efficiency

mainly through peripheral optimizations while preserving standard attention operators. Metrics based on theoretical operations often reflect GPU-centric assumptions, whereas on neuromorphic hardware, communication and data movement dominate energy cost. Recent analyses (Fang et al., 2024) show that dense self-attention thus becomes a major bottleneck, motivating hardware-aware attention redesign.

## 3. Preliminary

### 3.1. Spiking Neuron Model

We adopt the discrete-time leaky integrate-and-fire (LIF) neuron with hard reset as the fundamental spiking unit. Its dynamics are governed by a stateful membrane potential $u^{(t)}$ that integrates input and decays over time:

$$u^{(t)} = \lambda\, u^{(t-1)} \left(1 - s^{(t-1)}\right) + I^{(t)} \qquad (1)$$

where $\lambda \in (0, 1)$ is the leak factor, $I^{(t)}$ is the synaptic input at step $t$, and $s^{(t-1)}$ is the output spike from the previous time step. The term $\left(1 - s^{(t-1)}\right)$ implements a **hard reset**: the membrane potential is reset to zero immediately after a spike is emitted. When $u^{(t)}$ exceeds a firing threshold $V_{\text{th}}$, an output spike $s^{(t)}$ is generated:

$$s^{(t)} = \Theta\left(u^{(t)} - V_{\text{th}}\right) \qquad (2)$$

with $\Theta(\cdot)$ denoting the Heaviside step function. Inputs are encoded as binary spike trains over $T$ time steps, forming spatio-temporal representations that inherently preserve precise timing information.

### 3.2. SpikingResformer Architecture (Baseline)

SpikingResformer serves as our baseline architecture. Its overall network structure, illustrated in Fig. 1(a), follows the typical multi-stage design of vision Transformers: an input sequence is first processed by a spiking patch embedding layer, then passed through several consecutive Spiking Residual Blocks, and finally classified by a spiking head. Within each block, the core component is the **Multi-Head Dual Spike Self-Attention (MDSSA)** module, the detailed mechanism of which is depicted in Fig. 1(b).

Let $\mathbf{X} \in \mathbb{R}^{C \times H \times W}$ denote the spike tensor input to the MDSSA module after LIF neuron processing. The value ($\mathbf{Q}$), key ($\mathbf{K}$), and query ($\mathbf{V}$) features are constructed as follows:

$$\mathbf{Q} = \mathbf{X}; \mathbf{K} = \text{ConvPatch}(\mathbf{X}); \mathbf{V} = \text{ConvPatch}(\mathbf{X}); \qquad (3)$$

where $\text{ConvPatch}(\cdot)$ is a $P \times P$ convolution with stride $P$, producing downsampled K and V tensors.

For multi-head attention, these tensors are reshaped and

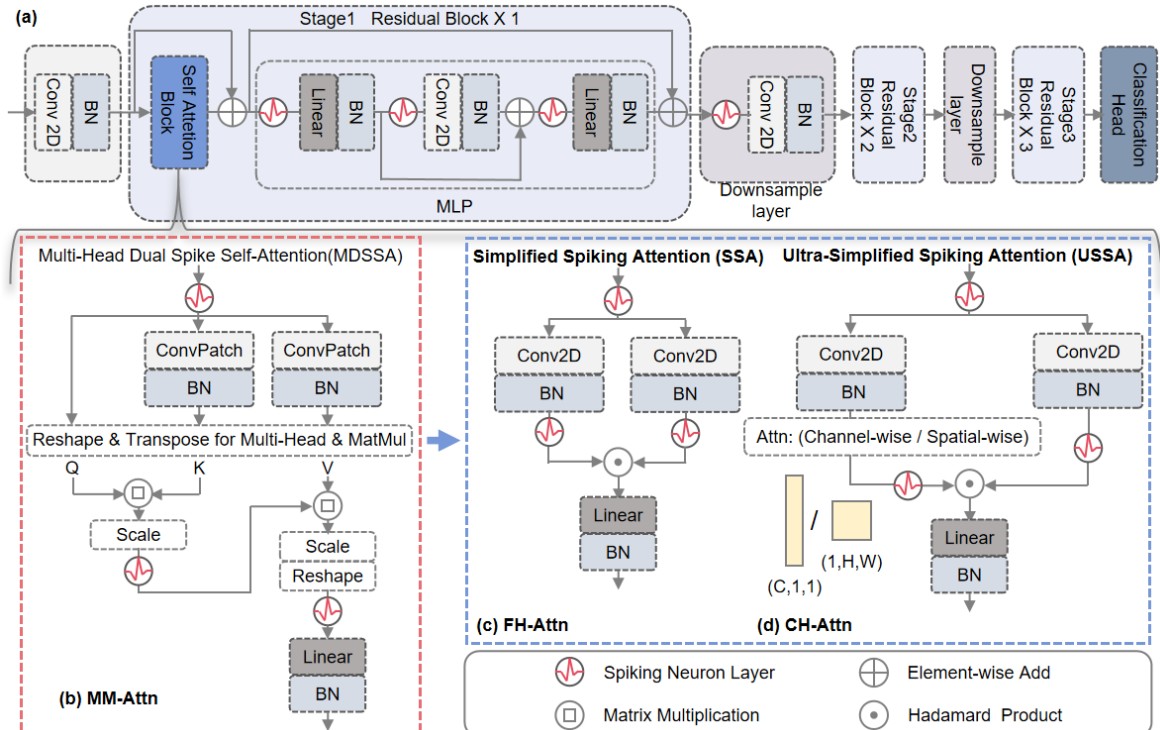

Figure 1. **Overall framework and attention module comparison.** (a) Overall architecture of the SpikingResformer-based network, where attention modules are embedded into residual blocks. (b) Multi-Head Dual Spiking Self-Attention (MDSSA), which relies on dense matrix multiplication, reshaping, transposition, and scaling operations. (c) Simplified Spiking Attention (SSA), which replaces matrix multiplication with full Hadamard-product-based spike interactions. (d) Ultra-Simplified Spiking Attention (USSA), which further compresses attention into channel-wise or spatial-wise gating to minimize communication overhead. Spiking neuron layers, matrix multiplication, element-wise addition, and Hadamard product are illustrated with corresponding symbols.

transposed as:

$$\mathbf{Q} \in \mathbb{R}^{M \times \frac{C}{M} \times HW}; \mathbf{K}^\top \in \mathbb{R}^{M \times \frac{HW}{P^2} \times \frac{C}{M}}; \mathbf{V} \in \mathbb{R}^{M \times \frac{C}{M} \times \frac{HW}{P^2}};$$
(4)

where $M$ is the number of attention heads, $H \cdot W$ and $\frac{H}{P} \cdot \frac{W}{P}$ represent the number of tokens before and after downsampling, respectively, and $\frac{C}{M}$ is the feature dimension per head.

The spiking attention output is computed via:

$$\mathbf{Attn} = \mathrm{LIF}(\mathbf{K}^\top \mathbf{Q}) \cdot \text{scale}, \tag{5}$$

$$\mathbf{Output} = \mathrm{LIF}(\mathbf{V}\, \mathbf{Attn}) \cdot \text{scale}, \tag{6}$$

where the scaling factor stabilizes attention magnitude.

Although MDSSA introduces self-attention into a spiking backbone, its dense all-to-all token interactions incur substantial communication and memory overhead, motivating the hardware-efficient attention simplifications proposed in this work.

### 3.3. Training Objective

The network is trained for classification using direct supervision on temporally aggregated outputs. Let $\mathbf{z}^{(t)} \in \mathbb{R}^C$

denote the output logits at time step $t$. We compute the time-averaged logits as

$$\bar{\mathbf{z}} = \frac{1}{T} \sum_{t=1}^{T} \mathbf{z}^{(t)} \tag{7}$$

where $T$ is the total number of simulation steps. The training objective is the standard cross-entropy loss

$$\mathcal{L} = -\log \frac{\exp(\bar{z}_y)}{\sum_{c=1}^{C} \exp(\bar{z}_c)} \tag{8}$$

with $y$ being the ground-truth label. Gradients are propagated through spiking neurons using surrogate gradients.

## 4. Method

This section details our proposed spiking attention framework. Section 4.1 introduces the overall architecture and formulates Simplified Spiking Attention (SSA) and Ultra-Simplified Spiking Attention (USSA) attention. Section 4.2 analyzes computational and communication complexity from dense to sparse attention. Section 4.3 examines the redundancy of spike masking and justifies a single-mask

simplification. Section 4.4 highlights the temporal benefits of early-spiking gating.

## 4.1. Overall Framework and Attention Formulation

Our method is developed on top of the SpikingResformer framework (Shi et al., 2024), which provides a residual hierarchical backbone equipped with spiking neuron dynamics, as illustrated in Fig. 1(a). We retain the original network architecture, stage-wise design, and training strategy, introducing two novel, hardware-oriented attention mechanisms—**Simplified Spiking Attention (SSA)** and **Ultra-Simplified Spiking Attention (USSA)**—to replace the standard attention module. This ensures that any observed differences in performance and efficiency are attributable solely to the proposed attention mechanisms.

We now detail the formulations of the proposed mechanisms. Let $\mathbf{X} \in \mathbb{R}^{C \times H \times W}$ denote the spike tensor after the first LIF neuron within the self-attention block, which is identical to the $\mathbf{X}$ defined in Eq. 4. In contrast to matrix-multiplication-based attention (MM-Attn) as instantiated in MDSSA, SSA and USSA remove explicit query–key matching and instead replace it with element-wise spike interactions that directly modulate token features.

**Simplified Spiking Attention (SSA).** SSA replaces the dense matrix-multiplication-based attention in standard MDSSA with an element-wise (Hadamard) product between spike features. Given $\mathbf{X}$, SSA first generates two spike feature branches through independent convolutional projections:

$$\mathbf{K} = \mathrm{LIF}(\mathrm{Conv}(\mathbf{X})), \quad \mathbf{V} = \mathrm{LIF}(\mathrm{Conv}(\mathbf{X})) \quad (9)$$

where $\mathrm{Conv}(\cdot)$ denotes a convolutional transformation and $\mathrm{LIF}(\cdot)$ applies spiking nonlinearity. The attention output is then computed via a full Hadamard product:

$$\mathbf{Y}_{\mathrm{SSA}} = \mathbf{K} \odot \mathbf{V}, \quad (10)$$

where $\odot$ denotes the Hadamard product. Compared with MDSSA, SSA completely removes dense matrix multiplication, token-wise projection, tensor transposition, and reshaping operations. This design preserves effective spatiotemporal modulation while significantly reducing computational and communication overhead, making SSA more suitable for deployment on neuromorphic hardware with strict energy constraints.

**Ultra-Simplified Spiking Attention (USSA).** USSA further simplifies SSA by compressing one attention operand into a broadcastable gating vector, thereby further reducing computational and communication overhead. Specifically, one branch is aggregated into a channel-wise or spatial-wise spiking gate:

$$\mathbf{g} = \mathrm{LIF}\left(\mathrm{Mean}(\mathrm{Conv}(\mathbf{X}))\right) \quad (11)$$

*Table 1.* Comparison of computation and communication complexity across attention paradigms.

| Paradigm | Computation | Communication |
|---|---|---|
| MM-Attn | $\mathcal{O}(N^2 D)$ | $\mathcal{O}(N^2)$ |
| FH-Attn | $\mathcal{O}(ND)$ | $\boldsymbol{\mathcal{O}(ND)}$ |
| CH-Attn | $\mathcal{O}(ND)$ | $\boldsymbol{\mathcal{O}(N)} / \boldsymbol{\mathcal{O}(D)}$ |

where $\mathrm{Mean}(\cdot)$ denotes dimensional averaging. For channel-wise attention, the mean operation is performed over the spatial dimensions ($H \times W$), yielding a descriptor $\mathbf{g} \in \mathbb{R}^{C \times 1 \times 1}$. For spatial-wise attention, the mean is taken over the channel dimension ($C$), resulting in a descriptor $\mathbf{g} \in \mathbb{R}^{1 \times H \times W}$, following the design in (Yao et al., 2023b).

The attention output is then computed via a Hadamard product:

$$\mathbf{Y}_{\mathrm{USSA}} = \mathbf{V} \odot \mathbf{g} \quad (12)$$

where the gating vector $\mathbf{g}$ is broadcast along the corresponding dimensions.

USSA represents the most hardware-efficient attention variant in our framework. By drastically reducing feature interaction scale, on-chip communication, and memory access, it achieves minimal implementation complexity, trading a certain degree of modeling capacity for substantial efficiency gains on energy-constrained neuromorphic hardware.

## 4.2. Theoretical Analysis: From Dense to Sparse Attention

SSA and USSA are designed to simplify dense matrix operations into sparse, element-wise computations. We consider three canonical attention paradigms—matrix-multiplication-based attention (MM-Attn), full Hadamard-product-based attention (FH-Attn), and compressed Hadamard-product-based attention (CH-Attn)—corresponding to MDSSA, SSA, and USSA (Fig. 1b–d). Treating them as abstract computational templates under a unified $N \times D$ tensor, we analyze computational and communication complexity, the latter capturing inter-token or inter-channel data movement that dominates energy on neuromorphic hardware.

**Definition 4.1** (Attention Paradigms). Let $N$ denote the number of tokens and $D$ the feature dimension.

(i) **MM-Attn:** $\mathbf{O} = (\mathbf{Q}\mathbf{K}^{\top})\mathbf{V}$, where $\mathbf{Q}, \mathbf{K}, \mathbf{V} \in \mathbb{R}^{N \times D}$.

(ii) **FH-Attn:** $\mathbf{O} = (\mathbf{Q} \odot \mathbf{K}) \odot \mathbf{V}$, where $\odot$ denotes element-wise multiplication. (This full form is theoretically analyzed here and later simplified to a single product in Section 4.3.)

(iii) **CH-Attn:** $\mathbf{O} = \mathbf{X} \odot \mathbf{g}$, where $\mathbf{g} \in \mathbb{R}^{N \times 1}$ or $\mathbb{R}^{1 \times D}$ is a broadcastable gating vector.

**Definition 4.2** (Communication Complexity). We consider

a token-wise distributed neuromorphic execution model, in which different tokens are mapped to distinct processing cores or memory blocks. Under this execution model, the communication complexity of an operator is defined as the number of inter-core or inter-memory-block data transfers required during inference, abstracted at the operator level. This abstraction aligns with common mesh-based or core-local neuromorphic architectures such as Loihi-like or SpiNNaker-like systems

This metric is particularly critical for neuromorphic systems, where data movement and spike routing typically dominate overall energy consumption.

**Proposition 4.3** (Communication Complexity of MM-Attn). *MM-Attn incurs $\mathcal{O}(N^2)$ communication complexity and $\mathcal{O}(N^2 D)$ computational complexity.*

*Proof.* Computing $\mathbf{QK}^\top$ requires each query token to interact with all key tokens. In a token-wise distributed model, this induces $\mathcal{O}(N^2)$ inter-core transfers. Each interaction is a $D$-dimensional inner product, giving $\mathcal{O}(N^2 D)$ computation. Multiplication with $\mathbf{V}$ follows the same dense pattern, preserving the asymptotic communication and computation complexity. $\square$

**Proposition 4.4** (Communication Complexity of FH-Attn). *FH-Attn incurs $\mathcal{O}(ND)$ communication complexity and $\mathcal{O}(ND)$ computational complexity.*

*Proof.* FH-Attn eliminates matrix multiplications and processes each token independently using element-wise operations. Under a token-wise distributed execution model, no inter-token interactions are required, thereby avoiding all-to-all communication. Each token accesses its own $D$-dimensional feature vector, resulting in $\mathcal{O}(ND)$ data movement across the memory hierarchy. The corresponding element-wise operations incur $\mathcal{O}(ND)$ computational complexity. $\square$

**Proposition 4.5** (Communication Complexity of CH-Attn). *Under a token-wise distributed neuromorphic execution model with broadcast-supported on-chip routing, CH-Attn reduces communication complexity to $\mathcal{O}(N)$ under token-wise gating or $\mathcal{O}(D)$ under channel-wise gating, while retaining $\mathcal{O}(ND)$ computational complexity.*

*Proof.* CH-Attn compresses one attention operand into a low-dimensional gating vector $\mathbf{g}$. Under token-wise gating, $\mathbf{g}$ is broadcast across tokens, incurring $\mathcal{O}(N)$ communication; under channel-wise gating, $\mathbf{g}$ is broadcast across channels, incurring $\mathcal{O}(D)$ communication. In both cases, dense token–token or feature–feature exchanges are avoided. The subsequent element-wise modulation of the input features involves $\mathcal{O}(ND)$ operations that can be executed using purely local data access. $\square$

As shown in Table 1, MM-Attn suffers from quadratic communication overhead, a fundamental hardware bottleneck. Hadamard-based paradigms (FH-Attn, CH-Attn) reduce communication to linear or sub-linear levels, aligning with sparse, distributed neuromorphic architectures and providing a theoretical foundation for efficient, hardware-feasible attention.

### 4.3. Redundancy Analysis of Spike Gating and Masking

Building on the Hadamard simplification, we further reduce data movement in SSA by replacing the double-masking structure $(Q \odot K) \odot V$ with a single, hardware-efficient mask, without loss of representational capacity.

**Proposition 4.6** (Redundancy of Double Masking on Spike Tensor $V$). *Let $Q, K, V$ be independent spike tensors whose elements follow Bernoulli distributions with firing rate $p$, i.e., $\mathbb{P}(X = 1) = p$ for $X \in \{Q, K, V\}$. Applying a second Hadamard-product mask on $V$ after $Q \odot K$ gating does not increase the information content and strictly reduces the entropy of the resulting spike representation for $0 < p < 1$.*

*Proof.* For single masking $Q \odot K$, the activation probability is $p^2$ with entropy $H(p^2)$. Double masking $(Q \odot K) \odot V$ gives $p^3$ with entropy $H(p^3)$. Since $0 < p < 0.5$, $H(p^3) < H(p^2)$, showing the second mask reduces information content without adding discriminative power. For a complete derivation, see Appendix A.

Empirical measurements on CIFAR-100 and DVS-Gesture show SSA spike rates $< 0.2$, satisfying $p < 0.5$ and confirming the redundancy of the second mask. Table 3 further validates that single-sided masking outperforms double masking in accuracy, reducing computation and communication while preserving information. $\square$

### 4.4. Temporal Advantage of Early-Spiking Gating

Having streamlined the computational structure to a single mask, we next optimize the temporal dynamics of the gating operation itself. Beyond static efficiency, the precise spike-timing—specifically, the choice between *early-spiking* and *late-spiking* gating—critically influences the temporal information available to the attention mechanism, thereby affecting final accuracy. This choice governs whether the gating signal is applied before or after the LIF neuron integrates its input over time, directly impacting the retention of historical context within the network's state.

**Proposition 4.7** (Temporal-State Preservation of Early- vs. Late-Spiking Gating). *Let $K(t), V(t)$ be input features at time $t$, and let $\mathrm{LIF}_1, \mathrm{LIF}_2$ denote leaky integrate-and-fire neurons. Compared to the conventional late-spiking gating*

$$Y_{late}(t) = \mathrm{LIF}_2\big(\mathrm{LIF}_1(K(t)) \odot V(t)\big) \qquad (13)$$

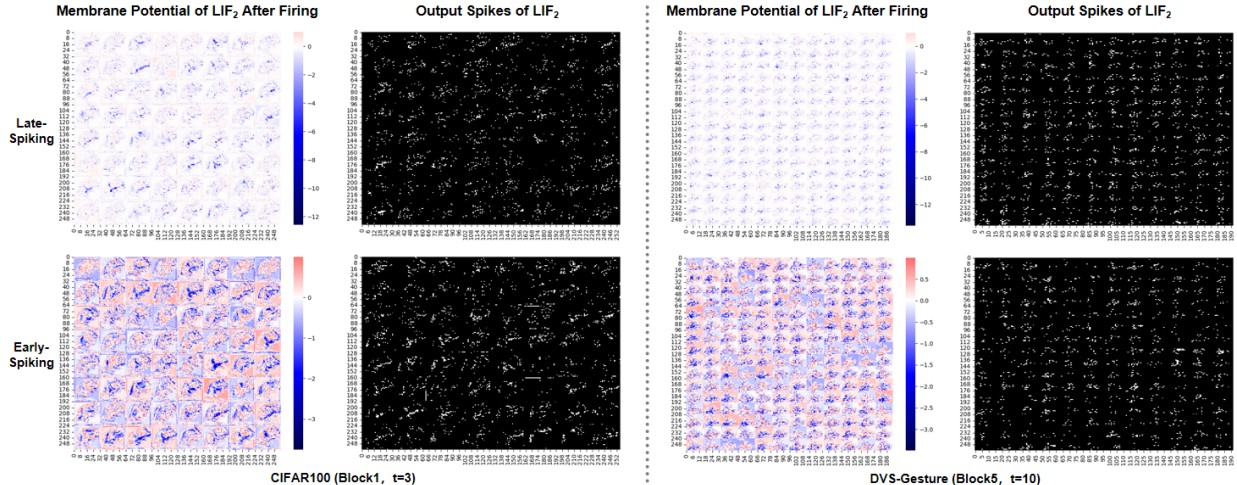

*Figure 2.* Under the SSA architecture, the post-activation membrane potentials and spike firing patterns of the $LIF_2$ neuron under early-spiking and late-spiking gating are compared in CIFAR-100 (first self-attention block, time step 3) and DVS-Gesture (fifth self-attention block, time step 10). Early-spiking gating preserves a denser membrane potential distribution without significantly increasing spike count, whereas late-spiking gating suppresses sub-threshold neural activity through pre-integration signal masking.

*Table 2.* Ablation study on simplifying matrix-multiplication-based attention (MM-Attn) to Hadamard-based attention (FH-Attn). "Patch/Conv." denotes the token embedding method. The fully simplified FH-Attn (bottom row), which removes both the scale factor and multi-head attention, serves as the prototype for the proposed Simplified Spiking Attention (SSA).

| Attention Type | Patch/Conv. | Scale | Multi-Head | CIFAR-100(%) | DVS-Gesture(%) |
|---|---|---|---|---|---|
| MM-Attn (Baseline) | ConvPatch | ✓ | ✓ | 78.15 | 95.83 |
| MM-Attn | 3×3 Conv | ✓ | ✓ | 77.37 | 94.79 |
| FH-Attn | 3×3 Conv | ✓ | ✓ | 75.91 | 93.40 |
| FH-Attn | 3×3 Conv | × | ✓ | 76.63 | 94.09 |
| FH-Attn | 3×3 Conv | × | × | 76.21 | 95.48 |

*the early-spiking gating employed in our design*

$$Y_{early}(t) = LIF_1(K(t)) \odot LIF_2(V(t)) \tag{14}$$

*preserves richer temporal information in the state of* $LIF_2$ *for* $t > 0$.

*Proof.* At $t = 0$, membrane potentials are initialized to zero; thus, both gating schemes produce identical outputs, as $LIF_1(K(0))$ induces the same binary mask. For $t > 0$, the two schemes diverge. In *late-spiking* gating, the mask $LIF_1(K(t))$ is applied *before* $LIF_2$, causing masked spatial positions to receive zero input. This prevents the accumulation of residual membrane potential across time and discards historical temporal context at those locations. In contrast, *early-spiking* gating allows $LIF_2(V(t))$ to integrate inputs independently over time, with the subsequent Hadamard product modulating only the output spikes while preserving internal membrane-potential states.

Fig. 2 compares membrane potentials and spike patterns under early- and late-spiking gating on CIFAR-100 (first self-attention block, time step 3) and DVS-Gesture (fifth self-attention block, time step 10). Early-spiking gating

maintains denser membrane-potential distributions without increasing spike counts, whereas late-spiking gating suppresses sub-threshold activity through pre-integration masking, leading to temporal information loss. □

## 5. Experiment

We evaluate the proposed SSA and USSA paradigms on both static and event-based vision benchmarks, with dataset details, training schedules, hyperparameters, and energy estimation methodology provided in Appendix B. Our experiments first assess architectural simplifications from dense matrix multiplication to Hadamard-product attention (Section 5.1), then analyze masking redundancy and the temporal impact of early-spike timing (Section 5.2), report theoretical energy consumption across attention designs (Section 5.4), compare SSA and USSA with state-of-the-art spiking neural networks (Section 5.5), further validate scalability on the large-scale ImageNet-1K dataset (Section 5.6, and evaluate hardware efficiency using a Lava-based Loihi-compatible simulation framework (Section 5.7).

## 5.1. Architectural Simplification: From Matrix Multiplication to Hadamard Product

We investigate the simplification of standard matrix-multiplication-based attention (MM-Attn) to the Hadamard-based FH-Attn paradigm. All branches replace spiking patch splitting with a dimension-preserving $3 \times 3$ convolution to ensure consistent spatial dimensions across Q, K, and V. Ablation studies evaluate the impact of removing the scale factor and multi-head attention. Table 2 shows that replacing patch splitting with convolution results in a minor performance drop (78.15% to 77.37% on CIFAR-100), while replacing MM-Attn with full Hadamard attention reduces accuracy to 75.91%, indicating that MM-Attn components are suboptimal for element-wise Hadamard attention. Removing the scale factor improves performance, and eliminating multi-head attention slightly boosts DVS-Gesture accuracy without affecting CIFAR-100. **The fully simplified FH-Attn (bottom row) thus forms the prototype for SSA,** replacing dense matrix multiplication and costly reshaping/transposition operations, and reducing computational complexity from $\mathcal{O}(N^2 D)$ to $\mathcal{O}(ND)$, making it more suitable for neuromorphic deployment.

## 5.2. Empirical Analysis of Masking Redundancy and Spike-Timing

This experiment investigates the redundancy of masks and the effect of spiking timing on classification accuracy in the Hadamard Attention paradigm. Based on the fully simplified FH-Attn (SSA prototype) from Table 2, we conducted two ablation studies: one on dual-mask configurations and another comparing early versus late spiking. The results are summarized in Table 3.

Across both datasets, dual-mask configurations yield lower accuracy than single-mask setups, confirming their redundancy (Proposition 4.6). Early spiking consistently achieves higher accuracy than late spiking (Proposition 4.7). The optimal SSA configuration—a single mask on the KV branch combined with early spiking—achieves 79.45% on CIFAR-100 and 97.56% on DVS-Gesture, outperforming the baseline MM-Attn by a clear margin.

## 5.3. From Simplified SSA to Compressed USSA

To improve the computational efficiency of SSA while maintaining competitive accuracy, we explore a compressed variant referred to as USSA. To isolate the effect of K-branch compression, the V-branch projection is fixed as a $3 \times 3$ convolution across all variants. We investigate two key factors: (1) the complexity of the K-branch projection, using either $3 \times 3$ or $1 \times 1$ convolutions, and (2) the compression strategy applied to the K-branch, implemented via channel-wise or spatial-wise reduction. These settings enable a systematic

*Table 3.* Ablation study on spike-gating timing and masking redundancy. "Late-Spiking" and "Early-Spiking" are defined in Eq. 13 and Eq. 14, respectively. "×" indicates that the corresponding branch is omitted. The last row corresponds to the final SSA configuration.

| QK Branch | KV Branch | CIFAR-100 | DVS-Gesture |
|---|---|---|---|
| Late-Spiking | Late-Spiking | 76.21% | 95.48% |
| Late-Spiking | × | 77.85% | 96.52% |
| × | Late-Spiking | 78.33% | 96.52% |
| Early-Spiking | Early-Spiking | 77.36% | 96.52% |
| Early-Spiking | × | 77.99% | 95.83% |
| × | **Early-Spiking** | **79.45%** | **97.56%** |

*Table 4.* Ablation study from SSA to USSA, evaluating the impact of K-branch projection and compression strategy. In the Compression (Comp.) column, Channel-Wise and Spatial-Wise reductions are denoted as Ch-W and Sp-W, respectively. All variants use a fixed $3 \times 3$ convolution for the V branch.

| Model | K-Branch | Comp. | CIFAR-100 | DVS-Gesture |
|---|---|---|---|---|
| SSA | $3 \times 3$ Conv | None | 79.45% | 97.56% |
| USSA | $3 \times 3$ Conv | Ch-W | 78.30% | 95.83% |
| | $3 \times 3$ Conv | Sp-W | 78.43% | 95.83% |
| | $1 \times 1$ **Conv** | **Ch-W** | **78.22%** | **95.48%** |
| | $1 \times 1$ Conv | Sp-W | 77.84% | 95.13% |

evaluation of the trade-off between model compression and task performance on both static (CIFAR-100) and event-based (DVS-Gesture) datasets.

Table 4 compares USSA variants with the uncompressed SSA baseline. All compressed variants exhibit a modest accuracy drop relative to SSA (79.45% on CIFAR-100 and 97.56% on DVS-Gesture), reflecting the expected compression–performance trade-off. While using a $3 \times 3$ convolution for the K branch yields slightly higher accuracy than a $1 \times 1$ convolution, the latter significantly reduces parameter count and computational cost. Among the compressed settings with $1 \times 1$ convolution, channel-wise reduction results in slightly higher accuracy than spatial-wise reduction. Considering the goal of low-power and hardware-efficient deployment, we adopt the $1 \times 1$ convolution with channel-wise compression as the default USSA configuration for subsequent experiments.

## 5.4. Energy Consumption Analysis

We evaluated the theoretical energy consumption of different self-attention designs—MDSSA* (baseline with ConvPatch removed), SSA, and USSA—on CIFAR-100 using the same SpikingResformer backbone. Fig. 3 shows per-block energy (mJ), and Table 1 reports arithmetic complexity. SSA and USSA reduce MDSSA's complexity from $O(N^2 D)$ (matrix-multiplication attention) to $O(ND)$ (full and compressed Hadamard attention), lowering estimated

*Table 5.* Comparison of SSA and USSA with state-of-the-art spiking SNNs on static (CIFAR-10/100) and event-based (DVS-Gesture) datasets. T denotes timesteps, and Acc. (%) denotes classification accuracy. Boldface indicates the best performance within each dataset.

| Method | Architecture | CIFAR-10 | | CIFAR-100 | | DVS-Gesture | |
|---|---|---|---|---|---|---|---|
| | | T | Acc. | T | Acc. | T | Acc. |
| *ResNet-based SNNs* | | | | | | | |
| TET (Zheng et al., 2021) | ResNet-19 | 4 | 94.44 | 4 | 74.47 | – | – |
| Dspike (Li et al., 2021) | ResNet-19 | 4 | 93.66 | 4 | 73.35 | – | – |
| STBP-tdBN (Zheng et al., 2021) | ResNet-19 | 4 | 92.92 | – | – | 40 | 96.87 |
| PSG (Wang et al., 2025) | ResNet-19 | 6 | 95.35 | 4 | 76.58 | 30 | **97.57** |
| *SpikingResformer-based Attention Models* | | | | | | | |
| STAttn (Lee et al., 2025) | SpikingResformer | 4 | 95.26 | 4 | 77.90 | – | – |
| MDSSA (Shi et al., 2024) (Baseline) | SpikingResformer | 4 | 95.89 | 4 | 78.15 | 16 | 95.83 |
| SSA (Ours) | SpikingResformer | 4 | **96.38** | 4 | **79.45** | 16 | 97.56 |
| USSA (Ours) | SpikingResformer | 4 | 95.80 | 4 | 78.22 | 16 | 95.48 |

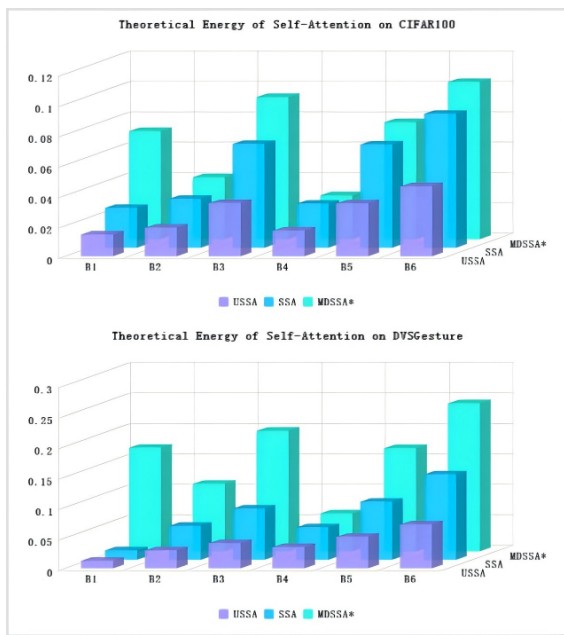

*Figure 3.* Theoretical energy consumption of different self-attention blocks in SpikingResformer on CIFAR-100 (top) and DVS-Gesture (bottom). The x-axis indicates the six attention blocks, the y-axis shows the attention variants (MDSSA*, SSA, and USSA), and the z-axis represents the estimated per-block energy in millijoules (mJ).

energy across blocks. However, the reductions remain modest, as the model accounts only for arithmetic operations and ignores communication costs such as tensor reshaping, transposition, memory access, and spike routing.

Under this model, the estimated per-image inference energy on CIFAR-100 is 1.16, 1.03, and 0.88 mJ for MDSSA*, SSA, and USSA, respectively; on DVS-Gesture, it is 4.36, 3.71, and 3.51 mJ. By substantially reducing communication complexity and eliminating data movement associated with dense attention, SSA and USSA are expected to achieve

greater energy savings on neuromorphic hardware. These results indicate that, for spiking self-attention models, practical efficiency depends more on minimizing communication overhead than on arithmetic operations alone.

### 5.5. Comparison with State-of-the-Art Methods

We compare the proposed SSA and its compressed variant (USSA) with representative state-of-the-art spiking neural networks, covering both convolutional SNNs and Transformer-based spiking architectures. As summarized in Table 5, the evaluation includes static image classification benchmarks (CIFAR-10 and CIFAR-100) and an event-based dataset (DVS-Gesture). For fair comparison, classification accuracy is reported under identical or comparable timesteps, and methods are grouped by backbone architecture (ResNet-based SNNs vs. SpikingResformer-based attention models) to isolate the effect of the attention mechanism itself.

SSA consistently achieves the highest accuracy among the compared methods across all datasets. On CIFAR-10 and CIFAR-100, it improves over the SpikingResformer baseline (MDSSA) by +0.49% and +1.30%, respectively, with identical timesteps. On the event-based DVS-Gesture benchmark, SSA reaches 97.56% accuracy, surpassing MDSSA and nearly matching PSG, which uses 30 timesteps. The compressed USSA variant demonstrates a favorable trade-off, maintaining competitive performance with only a modest decrease relative to SSA. Overall, these results validate the effectiveness and generality of the proposed Hadamard-based spiking attention mechanism across both static and neuromorphic vision tasks.

### 5.6. Results on ImageNet-1K

To further validate the scalability of the proposed methods in large-scale classification scenarios, we evaluate SSA and USSA on the ImageNet-1K dataset. As shown in Table 6,

*Table 6.* Comparison with state-of-the-art spiking neural networks on ImageNet-1K.

| Model | Params (M) | Acc. |
|---|---|---|
| Spikformer (Zhou et al., 2023) | 66.34 | 74.81 |
| Spike-driven Transformer (Yao et al., 2023a) | 36.01 | 74.66 |
| Spikingformer (Zhou et al., 2026) | 66.34 | 75.85 |
| Meta-SpikeFormer (Yao et al., 2024) | 32.80 | 77.20 |
| **SSA (Ours)** | 66.80 | 76.91 |
| **USSA (Ours)** | **50.26** | **77.27** |

SSA achieves 76.91% top-1 accuracy, outperforming prior SNN-based Transformer architectures such as Spikformer, Spike-driven Transformer, and Spikingformer. Moreover, the proposed USSA further improves the performance to 77.27% while reducing the parameter count from 66.80M to 50.26M, demonstrating a favorable trade-off between model capacity and recognition accuracy.

These results indicate that the proposed framework generalizes effectively to large-scale visual recognition tasks and maintains competitive scalability under high-category settings.

### 5.7. Hardware Efficiency Validation on Lava Simulation

To evaluate hardware efficiency, we adopt a Lava-based Loihi-compatible simulator following the energy model of Davies et al. (2018). On the DVS-Gesture dataset, we compare different attention mechanisms on the first self-attention module. As shown in Table 7, MM-Attn relies on token-wise mapping, which incurs frequent cross-token communication and transpose operations, consuming 3055.49 $\mu$J. In contrast, SSA and USSA use channel-wise mapping, avoiding explicit token mixing, and consume only 538.32 $\mu$J and 350.03 $\mu$J, respectively, consistent with their $\mathcal{O}(ND)$ and $\mathcal{O}(D)$ communication complexity analyses. All methods are evaluated under identical timestep settings. Real-hardware deployment is left as future work.

## 6. Conclusion

To address the deployment challenges of Transformer-SNNs on neuromorphic hardware, we propose hardware-friendly attention mechanisms—SSA and USSA—that replace matrix multiplication with Hadamard products, eliminating multi-head attention, scaling, and patch operations. This significantly reduces computational and communication overhead. Experiments on CIFAR-10/100, DVS-Gesture, and

*Table 7.* Energy comparison on Loihi-compatible Lava simulation.

| Attention Mechanism | Energy ($\mu$J) | Reduction |
|---|---|---|
| Matrix-Multiplication | 3055.49 | – |
| SSA | 538.32 | 82.4% |
| **USSA** | **350.03** | **88.5%** |

ImageNet-1K demonstrate that SSA outperforms existing methods on static and event-based benchmarks, while USSA delivers competitive accuracy with further efficiency gains, offering a practical path to deploy high-performance SNNs on energy-constrained neuromorphic chips.

## Acknowledgements

This work was supported in part by the National Natural Science Foundation of China (NSFC) under Grant Nos. 62476035, U24B20140, and 62306274; the Young Elite Scientists Sponsorship Program by CAST under Grant No. 2024QNRC001; and the China Postdoctoral Science Foundation under Grant Nos. GZB20250394, 2025T180427, and 2025M771543.

## Impact Statement

This paper advances efficient spiking neural networks for neuromorphic computing. Our SSA and USSA attention mechanisms reduce computational and communication complexity, enabling lower-power deployment of Transformer-based SNNs. This work is primarily methodological and introduces no specific negative societal impact.

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

# A. Proof of Proposition 4.6

*Proof.* We analyze the information content of spike representations under single and double masking.

For single masking via $Q \odot K$, since $Q$ and $K$ are independent, the probability that the masked spike is active is

$$\mathbb{P}(Q \odot K = 1) = \mathbb{P}(Q = 1)\mathbb{P}(K = 1) = p^2 \tag{15}$$

The corresponding entropy is

$$H(Q \odot K) = -p^2 \log p^2 - (1 - p^2) \log(1 - p^2) = H(p^2) \tag{16}$$

Similarly, for double masking via $(Q \odot K) \odot V$, independence implies

$$\mathbb{P}((Q \odot K) \odot V = 1) = p^3 \tag{17}$$

with entropy

$$H((Q \odot K) \odot V) = -p^3 \log p^3 - (1 - p^3) \log(1 - p^3) = H(p^3) \tag{18}$$

Since $0 < p < 1$, we have $p^3 < p^2$, and the Bernoulli entropy function $H(p)$ is increasing for $p \in (0, 0.5]$ and symmetric about $p = 0.5$, implying

$$H(p^3) < H(p^2) \tag{19}$$

Therefore, the second masking strictly reduces the information content without introducing additional discriminative structure, rendering it redundant. □

# B. Datasets, Implementation Details, and Energy Estimation

## B.1. Datasets

We evaluate the proposed SSA and USSA paradigms on both static and event-based vision benchmarks.

*CIFAR-10* (Krizhevsky et al., 2009) is a static image classification dataset consisting of 60,000 RGB images of size $32 \times 32$ across 10 object categories. It contains 50,000 training samples and 10,000 test samples, with 3 input channels.

*CIFAR-100* (Krizhevsky et al., 2009) shares the same resolution and data split as CIFAR-10 but contains 100 classes. Each image is of size $32 \times 32$ with 3 channels, comprising 50,000 training and 10,000 test samples.

*DVS-Gesture* (Gallego et al., 2022) (DVS128-Gesture) is a real-world neuromorphic dataset for gesture recognition, captured by a $128 \times 128$ DVS camera. It consists of 11 gesture classes, with 1,176 training samples and 288 test samples, each represented as asynchronous event streams with 2 polarity channels.

*ImageNet-1K* (Deng et al., 2009) is a large-scale static image classification dataset consisting of approximately 1.28 million training images and 50,000 validation images across 1,000 object categories. Each image is of variable size and is typically resized to $224 \times 224$ for model training and evaluation, with 3 RGB channels.

## B.2. Implementation Details

All experiments are conducted on a single NVIDIA A6000 GPU. Training schedules are: CIFAR-10 for 600 epochs, CIFAR-100 for 300 epochs, DVS128-Gesture for 300 epochs. We employ the AdamW optimizer, with a fixed neuron threshold of 1.0. The learning rate is set to $5 \times 10^{-4}$ for the first two datasets and $1 \times 10^{-4}$ for DVS128-Gesture. Other hyperparameters follow standard SNN training practices to ensure fair comparisons.

## B.3. Energy Consumption Estimation

Following prior work (Horowitz, 2014; Yao et al., 2023b), we adopt a standard analytical energy model assuming a 32-bit floating-point implementation in 45 nm CMOS technology. Under this model, a multiply-accumulate (MAC) operation consumes $E_{\mathrm{MAC}} = 4.6\,\mathrm{pJ}$, while an accumulate-only (AC) operation consumes $E_{\mathrm{AC}} = 0.9\,\mathrm{pJ}$.

**Baseline MDSSA.** In the baseline SpikingResformer with MDSSA, self-attention is implemented using matrix-multiplication-based operations. Due to spike-driven computation, attention-related matrix multiplications are modeled as accumulate-only (AC) operations rather than dense MACs. The first convolutional layer and the final fully-connected classification layer are implemented as dense MAC operations, while all intermediate convolutional layers and attention modules operate in a spike-driven accumulation manner.

Accordingly, the inference energy of MDSSA is estimated as

$$
\begin{aligned}
E_{\text{MDSSA}} = E_{\text{MAC}} \cdot \left( \text{FLOPs}_{\text{Conv}_1} + \text{FLOPs}_{\text{FC}_N} \right) \\
+ E_{\text{AC}} \cdot \left( \sum_{n=2}^{N-1} \text{FLOPs}_{\text{Conv}_n} \cdot fr_n + \text{FLOPs}_{\text{Attn-MM}} \cdot fr_{\text{attn}} \right),
\end{aligned}
\tag{20}
$$

where $\text{FLOPs}_{\text{Attn-MM}}$ denotes the operation count of matrix multiplications in the attention module (e.g., $QK^\top$ and $AV$), $N$ is the total number of convolutional layers, and $fr_n$ and $fr_{\text{attn}}$ denote the average spike firing rates of the $n$-th convolutional layer and attention module, respectively.

**SSA and USSA.** In contrast, the proposed SSA and USSA completely eliminate matrix multiplications from the attention module. Attention interaction is realized via early-spiking Hadamard-based gating, which effectively performs spike-wise binary masking. Such operations are functionally equivalent to logical AND and do not incur MAC or AC energy under the adopted analytical model, and its cost is negligible compared to accumulation-based operations. Therefore, attention-related accumulation energy is absent in SSA and USSA.

The inference energy of SSA and USSA is thus estimated as

$$
\begin{aligned}
E_{\text{SSA/USSA}} = E_{\text{MAC}} \cdot \left( \text{FLOPs}_{\text{Conv}_1} + \text{FLOPs}_{\text{FC}_N} \right) \\
+ E_{\text{AC}} \cdot \sum_{n=2}^{N-1} \text{FLOPs}_{\text{Conv}_n} \cdot fr_n.
\end{aligned}
\tag{21}
$$

**Discussion.** This formulation clarifies that the energy difference between MDSSA and SSA/USSA originates from the removal of attention-related accumulation operations rather than MACs. Nevertheless, since this analytical model accounts only for arithmetic computation, it does not capture communication, memory access, tensor reshaping, or spike routing overheads. These factors dominate energy consumption on neuromorphic hardware, explaining why theoretical energy reduction appears modest despite the substantial communication and structural simplifications achieved by SSA and USSA.

