# OpenReview forum: "Efficient Transformer Attention for SNNs via Hadamard Simplification"
_ICML.cc/2026/Conference — ICML 2026 regular_

### Official Review · Reviewer_LAa7 · 2026-02-19

**Soundness:** 3
**Presentation:** 3
**Significance:** 3
**Originality:** 3
**Overall Recommendation:** 3
**Confidence:** 2

**Summary:**

This paper targets a practical deployment bottleneck in Transformer-based Spiking Neural Networks (SNNs): standard self-attention requires dense matrix multiplications, reshaping/transposition, and global token interactions that are poorly matched to neuromorphic, event-driven execution, where data movement/communication often dominates energy. Building on a SpikingResformer-style backbone, the authors introduce two attention variants: Simplified Spiking Attention (SSA) (replacing matrix multiplication with a Hadamard-product interaction between two spike-feature branches) and Ultra-Simplified Spiking Attention (USSA) (compressing one operand into a broadcastable gate to further reduce communication).

The paper provides asymptotic analyses claiming a shift from O(N^2D) compute and O(N^2) communication (dense attention) to O(ND) compute with O(ND) or even O(N) communication (Hadamard / compressed gating), alongside two additional arguments: (i) “double masking” in Hadamard attention is redundant under a Bernoulli independence model and (ii) an “early-spiking” gating order preserves more temporal information in LIF states than “late-spiking” gating.

**Compliance With Llm Reviewing Policy:**

Affirmed.

**Key Questions For Authors:**

Can the authors report results on real hardware demonstrating real latency/energy gain? This would make the paper a good demonstration of a new method, which, otherwise, is only theoretical and lacks any guarantee of delivering real gains.

**Limitations:**

The biggest limitation is the missing deployment. Either you report it, or you clearly highlight it in introduction and conclusions.

**Strengths And Weaknesses:**

Strengths
- Clear hardware-motivated simplification path. The paper explicitly targets operations that tend to be inefficient on neuromorphic systems (dense token–token interactions, reshapes/transposes), and proposes attention forms that are closer to elementwise, local computation.
- Empirical gains on both static and event-based datasets. Reported improvements vs the baseline attention module are consistent across CIFAR-10/100 and DVS-Gesture, with SSA showing the best accuracy among the listed SpikingResformer-based attention variants.
- The early vs late gating analysis (and the associated membrane potential visualisation) is interesting and provides a good improvement.
- The paper acknowledges limitations of its energy model. The energy estimation explicitly notes that it does not capture communication/memory/routing overheads.

Weaknesses
- Several prior spiking-transformer works already pursue linear-complexity and/or multiplication-free / mask-based attention formulations as noted by the authors (e.g., Spikformer’s spiking self-attention; Spike-driven Transformer’s masking/addition-based attention with linear complexity; QKFormer’s linear spike-form attention variants). The paper should more precisely differentiate what is new (e.g., the exact SSA/USSA forms + the specific gating-order and redundancy analyses). Moreover, it should also compare in terms of accuracy with these methods.
- Hardware-efficiency claims are not validated on real neuromorphic platforms in the results. The paper primarily presents asymptotic communication complexity arguments and an analytical energy estimate; it contains no measurements on platforms such as Loihi-like or SpiNNaker-like systems, where routing and memory effects dominate.
- The token-wise distributed neuromorphic execution model is plausible as an abstract model, but neuromorphic mappings vary (token-wise vs channel-wise partitioning, routing fabric constraints, spike packet formats). The paper would be stronger if it showed that the stated traffic reductions correlate with measured traffic or energy on at least one concrete architecture.

---

> ### Author Rebuttal · Authors · 2026-03-30
>
> We thank the reviewer and hope the following clarifications address your concerns.
>
> 1. Novelty Clarification and Accuracy Comparison with Existing Spiking Attention Methods:
>
> We thank the reviewer for the valuable comments and will incorporate the following discussion into the revised manuscript. Prior works (e.g., Spike-driven Transformer, QKFormer) have explored using the Hadamard product to reduce computational complexity, but **they largely retain pairwise token interactions and focus primarily on computation optimization**. In contrast, **SSA/USSA are structurally reconfigured for neuromorphic hardware**: eliminating explicit token-to-token interactions enables local computation (SSA) and lightweight global sharing (USSA), reducing both computation and communication overhead. Importantly, **our approach avoids global reshaping operations (e.g., reshape, transpose) inherent to matrix multiplication, preserving spatiotemporal locality and asynchronous execution for event-driven systems**. Additionally, we remove scaling, redundant masking, multi-head mechanisms, and positional encoding, resulting in a more concise and hardware-friendly design.
>
> In comparison, some QKFormer-based methods (e.g., MSViT) adopt hybrid residual structures that combine a spike-based main branch with a floating-point side branch, introducing substantial multiply-accumulate operations. Since floating-point values dominate when combined with binary spikes, the model functionally relies on ANN-like floating-point computation, as supported by our ablation studies. In contrast, our design maintains a fully spike-driven paradigm and achieves competitive accuracy on benchmarks such as ImageNet-1K (see table below) with clear advantages in energy efficiency and structural simplicity. In summary, our core contribution is not a simple operator replacement but a co-optimized design of computation, communication, and architecture for neuromorphic hardware.
>
> **Results on ImageNet-1k**
>
> | Model | Param (M) | ACC (%) |
> |------|----------|---------|
> | Spikformer | 66.34 | 74.81 |
> | Spike-driven Transformer | 36.01 | 74.66 |
> | Spikingformer| 66.34 |75.85 |
> | Meta-SpikeFormer| 32.8 |77.2 |
> | QKFormer | 64.96 | 84.22 |
> | SSA (Ours) | 66.80 | 76.91 |
> | **USSA (Ours)** | **50.26** | **77.27** |
>
> 2. Limited Hardware Efficiency Validation and Lack of Real Neuromorphic Deployment:
>
> We agree that deployment on real neuromorphic hardware is essential for accurate evaluation of latency and energy consumption. Due to limited hardware accessibility, we have not yet achieved end-to-end deployment on physical Loihi or SpiNNaker systems; instead, we adopt a **Lava-based Loihi-compatible simulation framework** for energy modeling. The model is trained on NVIDIA using the DVS-Gesture dataset and then deployed to Lava, where execution on Loihi is simulated on a CPU backend. We focus on the first self-attention module to compare the energy consumption of conventional matrix multiplication with that of SSA/USSA.
>
> During simulation, we follow a token-wise mapping strategy and account for **NoC communication energy, synaptic spike operation energy, neuron update energy, and within-tile spike transmission energy**. All energy parameters are configured based on empirical measurements from “Loihi: A Neuromorphic Manycore Processor with On-Chip Learning.” **Under identical settings, the estimated energy consumption of SSA and USSA is 538.3240 μJ and 350.0316 μJ, respectively, while conventional matrix multiplication incurs a substantially higher energy cost of 3055.4859 μJ**, highlighting the clear energy efficiency advantages of the proposed methods. We will further discuss the limitations of simulation-based evaluation in the revised manuscript and identify real-hardware deployment as an important direction for future work.
>
> 3. Limited Empirical Link Between Communication Complexity and Concrete Neuromorphic Mappings:
>
> We thank the reviewer for this insightful comment clarifying the relationship between mapping strategies and energy efficiency. In our Lava simulation, **we adopt a token-wise mapping (one token per core)** to enable standardized energy accounting. Further investigation reveals that under this mapping, USSA does not achieve the expected energy advantage due to cross-core broadcast required by its global mean operation, introducing non-negligible NoC communication overhead. Based on this observation, **we argue that for SSA/USSA—designed to eliminate reshape operations and emphasize locality—a channel-wise mapping is more appropriate, enabling the theoretically derived communication complexities of O(ND) and O(D)**. In contrast, matrix multiplication-based attention aligns better with token-wise mapping. We will incorporate a detailed discussion of different mapping schemes in the revised manuscript to strengthen the connection between our method and concrete neuromorphic architectures. We thank the reviewer again for this valuable suggestion.

---

> > ### Author Rebuttal · Reviewer_LAa7 · 2026-04-07
> >
> > Thanks for the reply to my comments

---

> > > ### Author Response · Authors · 2026-04-07
> > >
> > > Thank you for your acknowledgement and for taking the time to review our rebuttal. Since you selected the option indicating that the concerns are “(c) Partially resolved or unresolved, but the remaining concerns are not easily addressed in a short rebuttal,” we would be very grateful if you could kindly provide more specific guidance on which issues you feel have not been fully addressed or which aspects may require more substantial revision. This would greatly help us better understand your concerns and further improve the paper; we will carefully take your comments into account and continue refining the manuscript accordingly.

---

### Official Review · Reviewer_Lz8A · 2026-03-02

**Soundness:** 2
**Presentation:** 3
**Significance:** 3
**Originality:** 2
**Overall Recommendation:** 4
**Confidence:** 4

**Summary:**

The contribution of this paper consists of the proposal and evaluation of two hardware-friendly attention mechanisms for spiking Transformer, i.e., Simplified Spiking Attention (SSA) and Ultra-Simplified Spiking Attention (USSA). The authors aim to replace the computationally intensive and communication-heavy matrix multiplications of standard Transformers with Hadamard products. Experiments on CIFAR-10/100 and DVS-Gesture demonstrate that these simplifications can maintain or even improve accuracy while significantly reducing computational and communication complexity.

**Compliance With Llm Reviewing Policy:**

Affirmed.

**Final Justification:**

My concerns about the novelty, large-scale datasets, hardware measurement, and code have been well-addressed.

**Key Questions For Authors:**

Compared to QKFormer or Spike-driven Transformer, which also utilize Hadamard products, what specific advantages do SSA and USSA have?

**Limitations:**

Please see the Weaknesses part. I will change my score if the authors can address all my concerns.

**Strengths And Weaknesses:**

**Strengths:**

1. The paper is clearly written and easy to understand.

2. The paper provides a complexity analysis showing a reduction from $\mathcal{O}(N^2D)$ to $\mathcal{O}(ND)$ of the proposed attention mechanisms.

**Weaknesses:**

1. The core improvement, i.e., using Hadamard products instead of matrix multiplications, has been utilized in previous studies. Thus, a detailed discussion should be provided about this mechanism.

2. The experiments are conducted on a small-scale dataset. More experiments on large-scale datasets like ImageNet are needed.

3. The paper emphasizes hardware friendliness, but the energy efficiency gains are based solely on the analytical estimations rather than empirical measurements on actual neuromorphic hardware.

4. The code is not available to check and reproduce.

---

> ### Author Rebuttal · Authors · 2026-03-30
>
> We thank the reviewer for the insightful comments and hope the following clarifications address your concerns.
>
> 1. Comparison with Existing Hadamard-Based Works:
>
> Thank you for your valuable feedback. We will incorporate the following discussion into the revised manuscript.
>
> We acknowledge that prior works (e.g., Spike-driven Transformer, QKFormer) have explored using the Hadamard product to replace matrix multiplication to reduce computational complexity. **However, these methods largely retain pairwise token interactions and primarily focus on computational optimization**. In contrast, **SSA/USSA are structurally reconfigured with neuromorphic hardware affinity in mind**: by eliminating explicit token-to-token interactions, they achieve local computation (SSA) and lightweight global sharing (USSA), thereby reducing both computational and communication overhead. **More importantly, our method avoids the global reshaping operations (e.g., reshape, transpose), thereby preserving spatiotemporal locality and asynchronous execution—key attributes for event-driven hardware**. In addition, we further streamline the architecture by removing scaling, redundant masking, multi-head mechanisms, and positional encoding, resulting in a simpler structure that maps efficiently to neuromorphic chips.
>
> By comparison, some QKFormer-based architectures (e.g., QKFormer, MSViT) achieve high accuracy but adopt a hybrid residual structure where the main branch is spike-based while the side branch remains floating-point; their outputs are summed and then fed into subsequent modules, introducing substantial multiply-accumulate operations. Moreover, since floating-point values dominate when summed with binary spikes, the overall architecture functionally relies on ANN-like floating-point computations to achieve accuracy, a point we have verified through corresponding ablation studies. In contrast, our design maintains a fully spike-driven paradigm throughout the network and validates its advantages via ablation and energy evaluation. **In summary, the core contribution of this work lies not in a simple operator replacement but in a co-optimized design of computation, communication, and architecture tailored to neuromorphic hardware constraints**.
>
> 2. On the validation of large-scale datasets:
>
> To verify the scalability and effectiveness of our method in large-scale scenarios, we conducted experiments on **ImageNet-1K** (1,000 classes). The results (see the table in our response) show that **SSA achieves a Top-1 accuracy of 76.91% with 66.80M parameters, while USSA attains 77.27% accuracy with only 50.26M parameters**, demonstrating superior parameter efficiency. Overall, our method maintains competitive performance on large-scale datasets and exhibits favorable scalability in terms of both accuracy and model size.
>
> **Results on ImageNet-1k**
>
> | Model | Param (M) | ACC (%) |
> |------|----------|---------|
> | Spikformer | 66.34 | 74.81 |
> | Spike-driven Transformer | 36.01 | 74.66 |
> | Spikingformer| 66.34 |75.85 |
> | Meta-SpikeFormer| 32.8 |77.2 |
> | SSA (Ours) | 66.80 | 76.91 |
> | **USSA (Ours)** | **50.26** | **77.27** |
>
> 3. On the lack of hardware measurement:
>
> We sincerely thank the reviewer for this insightful comment. We fully acknowledge that deployment on actual neuromorphic hardware is essential for accurately evaluating real-world latency and energy consumption. Due to limited hardware accessibility, end-to-end deployment on platforms such as Loihi or SpiNNaker is not yet feasible. As an alternative, we conduct **a hardware-oriented evaluation using the Lava framework with a Loihi-compatible CPU backend**, deploying model weights trained on NVIDIA hardware with the DVS-Gesture dataset for simulation.
>
> For a fair and controlled comparison, we analyze the energy consumption of the first self-attention module under a token-wise mapping strategy, comparing conventional matrix multiplication with SSA/USSA. The evaluation accounts for NoC communication, synaptic spike operations, neuron updates (active/inactive), and within-tile spike energy, with all parameters based on Loihi 1 measurements (Davies et al., IEEE Micro 2018, Table 2). **Under identical settings, conventional matrix multiplication consumes 3055.4859 μJ, while SSA and USSA consume 538.3240 μJ and 350.0316 μJ**, respectively, demonstrating clear energy efficiency advantages.
>
> 4. On code availability:
>
> We sincerely thank the reviewer for the valuable comments. To facilitate reproducibility and further evaluation, we have released our code in an anonymous repository (**https://anonymous.4open.science/r/Efficient-Transformer-Attention-for-SNNs-via-Hadamard-Simplification-06DF**), which includes the complete model architecture, training configurations, and Lava-based simulation scripts for hardware-oriented energy evaluation. We hope this resource supports the community in verifying and extending our work, and we are happy to provide additional details if needed.

---

> > ### Author Rebuttal · Reviewer_Lz8A · 2026-04-03
> >
> > My concerns have been addressed, and I am happy to raise my score.

---

> > > ### Author Response · Authors · 2026-04-04
> > >
> > > Thank you very much for acknowledging our revisions and for raising your score. We are very glad that your concerns have been fully addressed, and we truly appreciate your constructive and supportive feedback throughout the review process.

---

### Official Review · Reviewer_3JC9 · 2026-03-10

**Soundness:** 3
**Presentation:** 3
**Significance:** 3
**Originality:** 3
**Overall Recommendation:** 6
**Confidence:** 5

**Summary:**

To address the challenge of deploying Transformer-based SNNs on neuromorphic hardware due to dense computation and communication overhead, this paper proposes two simplified attention mechanisms, SSA and USSA, based on Hadamard products. Experimental results demonstrate that the proposed methods reduce both computational and communication complexity while achieving higher accuracy across multiple datasets.

**Compliance With Llm Reviewing Policy:**

Affirmed.

**Final Justification:**

Thank you to the authors for the clarification, particularly regarding the model’s performance on complex tasks. My concerns have been effectively addressed, and therefore I am inclined to raise my score.

**Key Questions For Authors:**

The experiments are conducted on CIFAR-10, CIFAR-100, and DVS-Gesture. Could the authors comment on how the proposed method might scale to larger datasets or more complex tasks such as ImageNet?

**Limitations:**

Yes

**Strengths And Weaknesses:**

# Strengths:
1.By replacing matrix multiplications in Transformer attention with Hadamard products, this paper proposes two attention mechanisms, SSA and USSA, which substantially reduce computational complexity and data communication overhead, making the Transformer architecture more efficient in SNNs.

2.Experiments on CIFAR-10, CIFAR-100, and DVS-Gesture demonstrate that the proposed method significantly improves computational efficiency while maintaining competitive classification accuracy, making Transformer-based SNNs more suitable for low-power neuromorphic hardware deployment.
# Weaknesses：
1.The paper primarily validates the proposed method on datasets such as CIFAR-10, CIFAR-100, and DVS-Gesture. While these results demonstrate its effectiveness, the absence of experiments on larger-scale datasets like ImageNet leaves the model's generalization capability on more complex tasks to be further verified.

2.Replacing matrix multiplication with Hadamard product may reduce global interactions between tokens. Does this simplification limit the expressive power of the attention mechanism?

3.The paper claims better suitability for neuromorphic hardware but does not include hardware experiments.

---

> ### Author Rebuttal · Authors · 2026-03-30
>
> We sincerely thank the reviewer for your careful evaluation of our work and valuable comments. Your suggestions have been instrumental in helping us improve the paper and clarify its contributions.
>
> 1. On the generalization to large-scale datasets:
>
>  We have added experiments on **ImageNet-1K** (1,000 classes) to validate the scalability and effectiveness of our method in large-scale settings. The experimental results are summarized in the table below. SSA achieves a Top-1 accuracy of 76.91% with 66.80M parameters, while USSA attains 77.27% accuracy with only 50.26M parameters, demonstrating superior parameter efficiency. These results indicate that our method maintains competitive performance on large-scale datasets and exhibits favorable scalability in terms of both accuracy and model size.
>
> **Results on ImageNet-1k**
>
> | Model | Param (M) | ACC (%) |
> |------|----------|---------|
> | Spikformer | 66.34 | 74.81 |
> | Spike-driven Transformer | 36.01 | 74.66 |
> | Spikingformer| 66.34 |75.85 |
> | Meta-SpikeFormer| 32.8 |77.2 |
> | SSA (Ours) | 66.80 | 76.91 |
> | **USSA (Ours)** | **50.26** | **77.27** |
>
> 2. On the lack of hardware validation:
>
> We thank the reviewer for raising this concern. We recognize the importance of hardware validation and have accordingly supplemented our work with a neuromorphic hardware-oriented evaluation. Specifically, we built a Loihi-compatible simulation environment using the CPU backend of the **Lava** framework. The model was trained on the NVIDIA platform using the DVS-Gesture dataset, after which the trained weights were deployed into Lava to simulate execution on Loihi. For a fair and controlled comparison, we selected the first self-attention module and performed statistical energy consumption analysis for conventional matrix multiplication versus the proposed SSA/USSA. During simulation, we strictly adhered to a token-wise mapping strategy and systematically accounted for NoC communication energy, energy per synaptic spike op, neuron update energy (differentiating between active and inactive states), and within-tile spike energy. All energy parameters were configured based on the measured data from Loihi 1 (Davies et al., IEEE Micro 2018, Table.2) to ensure high hardware consistency and credibility of the evaluation results. Experimental results show that, under identical settings, the estimated energy consumption of **SSA is 538.3240 μJ** and that of **USSA is 350.0316 μJ**, while **conventional matrix multiplication incurs a substantially higher energy cost of 3055.4859 μJ**, demonstrating the clear energy efficiency advantage of the proposed methods.
>
> 3. On the trade-off between global interaction capacity and representational power:
>
> We thank the reviewer for this important question. We agree that simply replacing matrix multiplication with the Hadamard product may reduce explicit token-to-token interaction. However, SSA/USSA are not simple substitutions; they compensate through structural reconfiguration. **SSA models local dependencies via a local receptive field, while USSA introduces a lightweight global sharing mechanism to enable cross-token information aggregation**, thereby preserving critical global context while reducing computational and communication overhead. More importantly, matrix multiplication relies on global reshaping operations (e.g., reshape), which disrupt spatiotemporal locality and introduce synchronization and buffering overhead, hindering the asynchronicity and event-driven nature of neuromorphic systems. In contrast, our method avoids such operations, with advantages validated by Loihi-based energy simulations using the Lava framework and ImageNet-1K experiments.
>
> Finally, we sincerely thank the reviewer again for your rigor and guidance. We hope our responses address your concerns and remain willing to further improve the paper in subsequent review rounds.

---

> > ### Author Rebuttal · Reviewer_3JC9 · 2026-04-03
> >
> > Thank you to the authors for the clarification, particularly regarding the model’s performance on complex tasks. My concerns have been effectively addressed, and therefore I am inclined to raise my score.

---

> > > ### Author Response · Authors · 2026-04-04
> > >
> > > Thank you so much for your acknowledgment and for increasing your score. We are glad that our clarifications, particularly on the model’s performance in complex tasks, have fully addressed your concerns. Your constructive feedback is much appreciated.

---

### Official Review · Reviewer_3sRt · 2026-03-11

**Soundness:** 3
**Presentation:** 3
**Significance:** 4
**Originality:** 2
**Overall Recommendation:** 5
**Confidence:** 4

**Summary:**

This paper proposes Simplified Spiking Attention (SSA) and Ultra-Simplified Spiking Attention (USSA), two attention mechanisms designed to improve the efficiency of Transformer-based spiking neural networks (SNNs) for deployment on neuromorphic hardware. The authors argue that existing Transformer-style SNN architectures inherit dense operations from artificial neural network (ANN) Transformers such as matrix multiplications, multi-head attention, and scaling which are poorly aligned with the neuromorphic hardware.

To address this mismatch, the paper replaces matrix multiplication in the attention mechanism with Hadamard products, thereby reducing both computational and communication complexity. The authors also analyze two additional aspects of spiking attention mechanisms: (1) redundancy in consecutive masking operations and (2) a spiking-order effect, suggesting that earlier spikes carry richer temporal information for attention modulation.

The proposed methods are evaluated on CIFAR-10, CIFAR-100, and DVS-Gesture, where SSA and USSA achieve competitive accuracy while significantly reducing theoretical computational and communication costs. The results suggest that simplified attention mechanisms may enable more hardware-friendly Transformer-based SNN architectures.

**Compliance With Llm Reviewing Policy:**

Affirmed.

**Final Justification:**

Thank you again to the authors for their rebuttal and clear answers to my questions, as well as including additional material to address these concerns. In my view this paper represents a novel concept which is well presented and investigated in depth. My only remaining concern is that the novelty/originality is incremental, however I believe this is a paper that is worthy of publication in ICML and is contributing to the field. Therefore I retain my recommendation of 5: accept

**Key Questions For Authors:**

1. Since the primary motivation is neuromorphic hardware efficiency, have the authors evaluated the proposed methods on actual neuromorphic platforms (e.g., Loihi, TrueNorth, or similar simulators)? Such results would significantly strengthen the paper.

2. How well does the proposed attention mechanism perform on tasks beyond image classification, such as event-based detection, segmentation, or temporal prediction?

3. Several recent efficient Transformer architectures propose linear or sparse attention mechanisms. How does SSA compare with these approaches in both performance and computational efficiency?

4. The paper claims that early spikes contain richer temporal information. Can the authors provide deeper theoretical justification or additional empirical analysis for this claim?

**Limitations:**

The paper discusses computational efficiency and masking redundancy but could include additional discussion of the lack of validation on real neuromorphic hardware and potential trade-offs between accuracy and efficiency in larger-scale models.

**Strengths And Weaknesses:**

Strengths

-The paper addresses a well-recognized issue in neuromorphic computing: the mismatch between ANN-inspired architectures and neuromorphic hardware constraints. Transformer-based SNNs often inherit operations that are inefficient for event-driven architectures, making the problem practically relevant.

-The core ideaofreplacing matrix multiplications with Hadamard products is conceptually simple and aligns well with the sparsity properties of SNNs.

-The paper provides a clear comparison of computational and communication complexity between standard attention mechanisms and the proposed methods. Demonstrating reductions from quadratic to linear complexity is an important result for hardware scalability.

-The experiments include both conventional vision datasets (CIFAR-10/100) and an event-based dataset (DVS-Gesture). This is appropriate given the neuromorphic focus of the work.

Weaknesses

-The core technical idea of replacing matrix multiplications with Hadamard-product-based attention mechanisms has already been explored in several prior SNN Transformer variants (for instance Lin, Xinxu, et al. "Spike-slr: an energy-efficient parallel spiking transformer for event-based sign language recognition." 35th British Machine Vision Conference 2024. 2024.). While the paper introduces refinements (SSA and USSA), the conceptual novelty appears incremental relative to existing work on simplified attention in SNNs.

- The experiments are relatively narrow in scope. Only three datasets are evaluated, and all tasks involve image classification. Additional tasks  would strengthen the claim that the method generalizes across applications (for instance from the following list of datasets  https://github.com/neuromorphicsystems/land/tree/main/datasets)

- Although the primary motivation is neuromorphic hardware efficiency, the paper reports only theoretical reductions in computational and communication complexity. No experiments are conducted on actual neuromorphic hardware platforms or energy simulators.

---

> ### Author Rebuttal · Authors · 2026-03-30
>
> We sincerely thank the reviewer for the encouraging comments, which are highly motivating and invaluable to our work.
>
> 1. Regarding the novelty and efficiency of SSA/USSA:
>
> Although prior works (e.g., Spike-SLR by Lin et al. ) have explored replacing matrix multiplication with Hadamard products, these methods still retain pairwise token interactions. In contrast, our SSA/USSA structurally redesigns the attention mechanism by eliminating explicit token-to-token interactions, implementing local computation (SSA) and lightweight global sharing (USSA), respectively. This design not only reduces computational and communication overhead but also avoids reliance on global reshaping operations, which often disrupt spatiotemporal locality and introduce extra synchronization and buffering, thereby affecting the event-driven characteristics of neuromorphic systems. We have supplemented comparisons with other methods on ImageNet-1K, summarized in the table. We will actively incorporate the reviewer’s suggestions to further strengthen the discussion and comparison with existing linear attention mechanisms.
>
> **Results on ImageNet-1k**
>
> | Model | Param (M) | ACC (%) |
> |------|----------|---------|
> | Spikformer | 66.34 | 74.81 |
> | Spike-driven Transformer | 36.01 | 74.66 |
> | Spikingformer| 66.34 |75.85 |
> | Meta-SpikeFormer| 32.8 |77.2 |
> | SSA (Ours) | 66.80 | 76.91 |
> | **USSA (Ours)** | **50.26** | **77.27** |
>
> 2. On the generalization to multi-task scenarios:
>
> We have added experiments on **ImageNet-1K** (1,000 classes) to validate the scalability and effectiveness of our method in large-scale settings. We agree that multi-task evaluation is important for further demonstrating model generalization; however, constrained by the time and space limitations of the rebuttal stage, we are currently unable to extend to additional tasks. Nevertheless, the design of SSA/USSA is inherently task-agnostic and offers strong generality. We have identified cross-modal tasks such as DVS-based action recognition and object detection as important future directions to further verify their applicability across scenarios.
>
> 3. On the limitation of hardware validation:
>
> We have supplemented the evaluation with a neuromorphic hardware-oriented assessment by building a Loihi-compatible simulation environment using the CPU backend of the Lava framework. The model was trained on the NVIDIA platform using the DVS-Gesture dataset, and the trained weights were deployed into **Lava** to simulate execution on Loihi. For a fair and controlled comparison, we selected the first self-attention module and conducted a statistical analysis of the energy consumption of conventional matrix multiplication versus the proposed SSA/USSA. During simulation, we strictly followed a token-wise mapping strategy and systematically accounted for NoC communication energy, energy per synaptic spike op, neuron update energy (differentiating between active and inactive states), and within-tile spike energy. All energy parameters were configured based on the measured data from Loihi 1 (Davies et al., IEEE Micro 2018, Table.2) to ensure good hardware consistency and credibility of the evaluation results. Experimental results show that, under identical settings, **the estimated energy consumption of SSA is 538.3240 μJ and that of USSA is 350.0316 μJ, while conventional matrix multiplication incurs a substantially higher energy cost of 3055.4859 μJ**, demonstrating the clear energy efficiency advantage of the proposed methods.
>
> 4. On the theoretical basis and empirical analysis of the temporal information richness of early spikes:
>
> We thank the reviewer for the insightful question. Although early-spiking and late-spiking are operationally equivalent (masking vs. dynamic integration), the timing of applying the mask—which serves as an irreversible information bottleneck—is crucial: introducing constraints earlier is more likely to disrupt the original temporal structure, leading to greater information loss. **To investigate this, we additionally computed the information entropy of the outputs from both mechanisms**. **On CIFAR-100, early-spiking consistently yields higher average information entropy (0.0250) than late-spiking (0.0235) across modules**, indicating that it retains richer temporal information. **On DVS-Gesture, the information entropy for both is 0.0304**; we speculate that this is due to the stronger temporal redundancy inherent in DVS data, which is also consistent with the instability of the ablation results for this dataset in Table 3. We appreciate the reviewer’s in-depth discussion, and we will add this analysis to the revised manuscript.
>
> Finally, we sincerely thank the reviewer again for your rigor and thoroughness. We hope the above responses have addressed your concerns. Should there remain any shortcomings in the manuscript, we would be glad to continue receiving your guidance and further improve the work in the next review round.

---

> > ### Author Rebuttal · Reviewer_3sRt · 2026-04-06
> >
> > Thank you to the authors for their rebuttal, which has addressed all the main concerns I had raised

---

### Decision · Program_Chairs · 2026-04-30

**Decision:**

Accept (regular)

**Comment:**

This paper studies hardware-friendly attention design for Transformer-based spiking neural networks and proposes two simplified attention mechanisms, SSA and USSA, that reduce computation and communication by replacing standard matrix-multiplication attention with Hadamard-style interactions. The reviewers generally agree that the paper addresses a practically relevant problem for neuromorphic and event-driven computing, and that the proposed methods are clearly motivated and show encouraging empirical results on the reported benchmarks. The rebuttal further strengthens the paper by adding ImageNet-1K results, code availability, and a Lava/Loihi-compatible simulation-based energy analysis.

At the same time, reviewers raised several important concerns. In particular, multiple reviewers noted that the novelty is somewhat incremental relative to prior Hadamard-based or linear-complexity spiking attention designs, that the original empirical scope was limited, and that the hardware-efficiency claims were supported primarily by analysis rather than concrete deployment. The rebuttal addresses a number of these questions constructively, and several reviewers raised their scores.

After carefully reading the paper and rebuttal, I view this work primarily as an engineering-driven, hardware-oriented design refinement rather than a major conceptual advance. For a paper of this type, the empirical support carries particular weight. In my view, the original submission did not yet meet that standard: the empirical scope and scale were limited, and the hardware claims were supported mainly by complexity analysis. The rebuttal addresses several of these gaps constructively, but in doing so it brings the paper to a relatively solid level rather than to an unusually strong one as suggested by the most enthusiastic scores. Even after rebuttal, the large-scale evidence is encouraging but not yet as strong as that in previous spiking transformer works.

Overall, I find the paper to be a worthwhile contribution, but not as strong as some of the highest scores imply. Therefore, I recommend Weak Accept.